# Experts' views on translating NHS support to stop smoking in pregnancy into a comprehensive digital intervention

Lisa McDaid[1]*, Pippa Belderson[1], Joanne Emery[1], Tim Coleman[2], Jo Leonardi-Bee[3], Felix Naughton[1]

1 School of Health Sciences, University of East Anglia, Norwich, United Kingdom, 2 Division of Primary Care, School of Medicine, University of Nottingham, Nottingham, United Kingdom, 3 Centre for Evidence Based Healthcare, School of Medicine, University of Nottingham, United Kingdom

* l.mcdaid@uea.ac.uk

**Data Availability Statement:** The anonymised data that support the findings of this study are available on request from the study sponsor: Faculty of Medicine and Health Research and Innovation

## Abstract

Many pregnant smokers need support to quit successfully. In the United Kingdom, trained smoking cessation advisors deliver structured behavioural counselling alongside access to free nicotine replacement therapy (NRT); known as the 'Standard Treatment Programme' (STP). Pregnant smokers who access STP support are more likely to quit, but uptake is low. A digital intervention could be offered as an adjunct or alternative to existing STP support to increase cessation rates. However, there are few pregnancy-specific digital options routinely available and, among those that are, there is limited evidence of their effectiveness. This study investigated experts' views on the feasibility of translating the STP into a comprehensive digital intervention. Virtual group and individual interviews were undertaken with 37 experts (11 focus groups, 3 interviews) with a real-time voting activity in the focus groups to prompt discussion. Framework Analysis was applied to the data to examine themes and patterns. Experts were supportive of a digital translation of the STP and considered most behavioural counselling content to be transferable. However, replicating human-to-human accountability, empathy and the ability to go 'off-script' was thought more challenging. Suggestions for how this might be achieved included tailoring and personalisation, use of artificial intelligence tools, peer support and the option to escalate contact to a human advisor. Experts had mixed views on the role that exhaled breath carbon monoxide monitoring might have in a digital cessation intervention for pregnancy. Electronic provision of free NRT, and potentially e-cigarettes, without interpersonal support was generally well received. However, experts had concerns about it exacerbating low NRT adherence, governance issues (e.g. being accountable for the suitability of recommended products), and people's ability to misrepresent their eligibility. The STP was considered largely transferable to a digital intervention and potentially helpful for cessation in pregnancy, so merits further development and evaluation.

Services, rin.fmh@uea.ac.uk. The data are not publicly available due to ethics restrictions.

**Funding:** This article presents independent research funded by the National Institute for Health and Care Research (NIHR) under the Programme Development Grant scheme (NIHR203305). The funding was awarded to FN (Principal Investigator), TC & JLB (co-applicants), and LM, PB & JE received all or part of their salary from the funder. TC is an NIHR Senior Investigator. The funders had no role in study design, data collection and analysis, decision to publish, or preparation of the manuscript.

**Competing interests:** The authors have declared that no competing interests exist.

## Author summary

Smoking in pregnancy is a leading preventable cause of poor pregnancy outcomes and health problems for children. Evidence-based behavioural counselling and nicotine replacement therapy (NRT) interventions to support pregnant smokers are typically delivered by a trained advisor. In the United Kingdom, the recommended National Health Service intervention, known as the 'Standard Treatment Programme' (STP) for pregnancy, combines both counselling and NRT and has been shown to increase the likelihood of quitting success—yet uptake is low. In this study, we wanted to explore experts' views on the feasibility of delivering the STP digitally to reduce access barriers. Experts were supportive of a comprehensive digital smoking cessation intervention for pregnancy. Most components from the STP were considered transferable from human to machine-led support. However, various considerations were highlighted which could potentially impact on engagement with a digital intervention, such as a lack of human accountability and the need to address the broader contextual influences of smoking behaviour. Our study provides useful insights to support and shape the future development of digital cessation interventions for pregnancy.

## Background

Smoking in pregnancy is associated with numerous pregnancy complications and life-long conditions for babies [1–3]. While maternal smoking rates have declined in high-income countries [4], there are socio-demographic differences in who is more likely to smoke when becoming pregnant and who is more likely to quit. Deprivation is overwhelmingly associated with both, reinforcing health inequalities [5,6]. Along with living in more unfavourable financial and social conditions, psychological factors, dependence on tobacco and access to effective and acceptable support, can all interact to make quitting smoking in pregnancy more difficult [7]. The World Health Organisation (WHO) recommends interpersonal counselling as a first-line treatment for smoking cessation in pregnancy [8]. It has been shown to increase chances of quitting in late pregnancy by over 40% compared to usual care [9] and is typically more effective when offered in combination with other behavioural approaches, such as health education, feedback (e.g. carbon monoxide testing) and incentives [9–11].

In the United Kingdom (UK), clinical guidelines recommend the use of nicotine replacement therapy (NRT) alongside behavioural support from the earliest opportunity [12]. These guidelines are implemented through the 'Standard Treatment Programme' (STP) [13], an evidence-based intervention designed to be delivered by specialist stop smoking advisors. It consists of 1) a pre-quit assessment 2) setting a quit date, 3) weekly behavioural counselling sessions, 4) regular carbon monoxide (CO) monitoring, and 5) the provision of free NRT with encouragement to use two forms (patch plus fast-acting product). The weekly sessions are designed to address modifiable determinates of smoking at an individual level, such as enhancing motivation to quit, planning how to manage tempting situations and cravings, and addressing any NRT usage issues. Offering support to family alongside the STP is also recommended.

However, while many pregnant smokers' express interest in accessing specialist NHS cessation support, only around 10% take up the offer [14,15]. The reasons for this low uptake are likely to be complex; some pregnant smokers may not be ready to quit or not want to quit entirely [16], but almost half of pregnant smokers will make a quit attempt [17]. Other studies suggest that barriers to access exist, including time constraints (e.g. because of childcare and

work responsibilities) [16], a 'postcode lottery' in service availability and quality [16,18], finding services inflexible in terms of the treatment options offered [16,19], and fear of judgement by healthcare professionals [16,20,21]. In the UK, the NHS Long Term Plan smokefree pregnancy pathway [22] has recently been introduced to provide a more integrated system of maternity-led support, though it is unclear whether this will meet the support needs of those who find it hard or do not want to engage with standard interpersonal support as most barriers remain.

Increasing the range of digital health tools available is a top priority for the NHS [22] and may help to circumvent some of the reasons why pregnant smokers do not use interpersonal cessation support; for example, digital support can improve convenience and be accessed at any time [7,23,24], it can increase anonymity and reduce the fear of feeling judged [23,25], and it may be more attractive to underserved groups, such as young adults and teenagers [25]. A range of digital smoking cessation interventions are available for the general population in high income countries (e.g. websites, text messages, smartphone applications or 'apps'). These can deliver low-cost support, improve reach due to high scalability and can assist smoking cessation [26], especially in combination with pharmacotherapy [27]. However, few of these interventions are orientated towards pregnancy, which is the primary reason most pregnant smokers try to quit [28], and of those that are and have shown potential for cessation [29], which are mainly computer- and text-based, none have been routinely implemented in the NHS. There is also still much to learn about the best strategies to promote uptake and engagement with digital cessation interventions in this population, as this can impact on quitting success, and low engagement is a key reason why a previous cessation app protype for pregnancy was abandoned [30].

To date, no digital cessation interventions for pregnancy have tried to replicate the combination of behavioural support and NRT, as used in the STP. The safe, remote, provision of NRT, which can help to address nicotine withdrawal and likely increases smoking cessation during pregnancy [31], could further enhance the effectiveness of digital support. Similarly, while the safety of electronic cigarettes (e-cigarettes) in pregnancy remains unclear [32], there is a rapid emergence of evidence in this area, including a recent randomised control trial demonstrating their potential efficacy as a smoking cessation tool in pregnancy [33]. E-cigarette use is supported in the STP [13] and therefore it is important to improve understanding of how e-cigarettes could potentially be provided remotely to increase cessation should provision become standard care in the future.

This study investigates the views of smoking in pregnancy experts on the feasibility of translating the STP from interpersonal to digital delivery, including the remote provision of CO monitoring and nicotine substitution products. It also examines how a digital intervention might complement and be integrated within the pregnancy pathway and alongside the STP. By exploring additional modes of access to effective cessation support in pregnancy, this will hopefully lead to more successful quit attempts.

## Methods

### Study design

The study has pragmatic origins due to its practice-driven focus [34]. Qualitative research methods, consisting of online semi-structured focus groups and interviews, were used. The study was approved by Research Ethics Committees at the University of East Anglia (Project ID- ETH2223-0925). This study is reported using the Consolidated Criteria for Reporting Qualitative Research (COREQ) [35].

### Research team and public partners

The lead researchers (LM & PB) were female, have PhDs and experience in undertaking smoking cessation research. The wider research team (FN, JE, TC, & JLB) and our advisory group (see acknowledgements) also have considerable experience in the field and understand the complexities of stopping smoking in pregnancy. The research team worked closely with three public partners to develop participant materials, the topic guide (including prompts for potential features of a digital support package) and interpret the findings.

### Participants

Recruitment took place via email invitations sent out to individuals or organisations in the smoking in pregnancy field. Potential participants were identified from contacts known to the research team, through the National Centre for Smoking Cessation and Training (NCSCT), and snowballing methods. We grouped 'experts' into three broad categories: 1) *Academics*— who have published research on digital interventions for smoking cessation in pregnancy, 2) *Policy professionals*, *commissioners*, *and trainers*–in leading policy roles, responsible for developing or delivering specialist training, or commissioning specialist cessation services for pregnancy, and 3) *Practitioners*–responsible for coordinating or delivering specialist support to pregnant smokers, including stop smoking advisors and midwives. A purposive sampling approach was used with the aim to recruit a 1: 1: 2 (practitioners) allocation ratio. Most participants were recruited from the UK and Channel Islands, but international academics also participated to bring a broader perspective. Recruitment and data collection stopped after data saturation was reached, meaning no new data were arising [36].

### Procedure

Informed consent was obtained prospectively using an online form. Due to the virtual nature of the focus groups, the maximum number of participants per group was limited to five [37]. Overall, 11 focus groups took place. Seven had participants from a single expert category and five were mixed. The discussions were led by a facilitator and co-facilitator (LM & PB). A real-time voting activity using the online tool 'Mentimeter' [38] (see activity in supporting information S1 File) was used to prompt discussion on the transferability of standard treatment components. A semi-structured topic guide was developed to allow further exploration of the benefits and limitations of a digital cessation intervention for pregnancy (see supporting information S2 File). It also looked at the degree to which experts felt that it could be complementary to, or independent from, standard care. Focus groups lasted for approximately two hours. Reflective notes focusing on key themes were made afterwards. Three individual interviews were undertaken to accommodate participants who were unable to attend the focus groups. The interviews took a more flexible approach depending on participant availability and lasted between 20–54 minutes. Data collection took place between May 2022 and October 2022.

### Analysis

The focus groups and interviews were video recorded on Microsoft Teams, with live automatic transcription. Transcripts were corrected and anonymised and then uploaded to NVivo v12 software. The Framework Method [39,40] was used to analyse the data with cases organised by expert category. The initial thematic framework was jointly created by the researchers carrying out the analysis (LM, PB), using inductively- and deductively- derived themes. Initial coding was carried out by one researcher, with secondary coding performed on over half of the transcripts ($n$ = 9) by the second researcher. The preliminary research findings were subsequently

discussed with the wider research team, study advisory group and our public partners. A summary of the findings for comment was also sent to all participants. Feedback from this participant validation activity helped to further refine the findings and ensure they resonated with participants.

The mean score for each 'Mentimeter' rating of transferability was calculated (from 0 'very difficult to transfer' to 10 'very easy to transfer'). This was then used to categorise the transferability into tertiles: 'Hard' (m = 0–3.3), 'Medium' (m = 3.4–6.7), 'Easy' (m = 6.8–10). As this exercise was primarily to provide stimulus for the group discussion, participant's views sometimes evolved during the discussion in light of further reflection and new information.

## Results

Thirty-seven experts took part in the study, with time being the main barrier to participation. Experts worked across a range of settings and geographic locations. Details about the numbers in each expert category and work location are summarised Table 1.

Three main themes are discussed in this paper. These focus on 1) The transferability of interpersonal counselling to a digital intervention, 2) Remote digital provision of exhaled breath CO monitoring and nicotine substitution products, and 3) Integration and reach. Participant identifiers have been used for direct quotations. These consist of the focus group number, participant number, and the expert category code (AC = Academic, PTC = Policy professional, trainer, commissioner, PR = Practitioner).

### Theme 1: The transferability of interpersonal counselling to a digital intervention

Experts were supportive of a digital cessation intervention for pregnancy. There were thought to be few viable options to currently offer pregnant smokers who feel unable or are unwilling to engage with the STP. Geographic inequalities in the distribution and quality of pregnancy

**Table 1. Characteristics of participants.**

| Role | Number (%) (n = 37) |
| --- | --- |
| Health practitioner (including midwives and service managers) | 19 (51%) |
| Policy professional, trainer, commissioner | 10 (27%) |
| Academic | 8 (22%) |
| **Country and Region** | |
| *England* | |
| National | 5 (14%) |
| East of England | 6 (16%) |
| London | 1 (3%) |
| Midlands | 7 (19%) |
| North East and Yorkshire | 3 (8%) |
| North West | 3 (8%) |
| South East | 1 (3%) |
| South West | 3 (8%) |
| *Elsewhere* | |
| Wales | 2 (5%) |
| Scotland | 3 (8%) |
| Channel Islands | 1 (3%) |
| International | 2 (5%) |

cessation services were also noted. Table 2 summarises experts' views on the transferability of STP components and how they might be delivered digitally.

**Assess personal context.** Almost all participants thought that it would be straightforward to learn about the pregnancy and capture information on smoking history, readiness to quit and physiological and mental functioning using digital methods ("*. . .doing the initial assessment it's quite paint by numbers. It's questions that would have a fairly standard answer.*"-FG9-28-PR). However, it was recognised that this approach would perhaps lose some of the "*nuance and complexity*" (FG2-06-AC) of human interaction, such as understanding how past experiences of quitting might influence a current quit attempt ("*To me this question is probably one of the most important questions. Assessing what someone's done before and exploring what didn't work [. . .] that sort of motivational dialogue to draw that information out.*"-FG9-27-PR). It was also thought that broader contextual factors which can drive tobacco addiction would be difficult to address in a digital intervention.

In several of the focus groups, the topic of self-reporting smoking behaviour led to a discussion about honest responding and how this might be influenced by digital methods. Responses were mixed, with some participants suggesting that the anonymity of digital support might make people feel less judged ("*One of my feelings is people might actually be more accurate [. . .] when they're trying to describe their smoking habit just to a physical person, [people] tend to massively underreport.*"-FG1-02-AC). In contrast, others felt that pregnant smokers may not always be honest with themselves and it requires investigative questioning to find out exactly what is going on ("*I mean you can ask them 'Do you smoke in the house?' and they'll say 'No' and then they'll go 'Well only in the kitchen' or 'Only when the children have gone to bed at night' but they've said 'No'*"-FG11-35-PR).

**Relationship and rapport.** Relationship and rapport were considered the most challenging aspects of the STP to replicate digitally, particularly if there was no human component within the intervention. However, not all participants saw this as problematic. It was suggested that some people prefer autonomy when it comes to a quit attempt and not all relationships between health professionals and pregnant smokers are positive, which can have a detrimental effect on the quit attempt ("*I think when the relationship works it really works well and that rapport is important, but the rapport doesn't always happen [. . .] You do hear some women saying 'I couldn't stand my midwife' or 'I didn't like my stop smoking advisor' and that's why we get so many DNAs [Did Not Attends]–that rapport is something desirable in our heads but it's not necessarily something that the women experience or even want, so having more automated information through an app could be preferable to some women. . .*"-FG4-09-PTR). However, others strongly believed that accountability to others and the perception of genuine empathy were a core part of a successful quit attempt ("*. . .it's one thing doing it to an app. i.e. yourself that nobody sees, it's another having to show it to another person and all of the evidence that we have is that this sort of sense of accountability is crucial to maintaining behaviour change effort. . .*"-IV2-37-AC, "*. . .it's really helpful when people feel as though they matter. . .*"-FG11-34-PR).

Various approaches were suggested to help build a sense of rapport and accountability into a digital intervention. These included tailoring and personalisation, having the flexibility to account for different quit journeys and peer support. Others, especially practitioners, thought there needed to be a human element to the digital intervention, whether this was embedded within the intervention itself or by linking with existing stop smoking provision. Some participants were also concerned about how easy it would be to disengage with a digital intervention ("*. . .it's easier to say no to an app or just delete it or turn it off or mute it than it is to say to a person. . .*"-FG7-19-PR). Interpersonal support was thought to be especially important for those who find it hard to quit ("*. . .trying to find why somebody is struggling really can take you round and round so many different areas and that's where that human conversation [. . .] it's the*

**Table 2. Summary of findings.**

| Component | NCSCT Clinical Checklist | Transferability | How | Pros | Cons |
|---|---|---|---|---|---|
| **BEHAVIOURAL COUNSELLING AND INFORMATION** | | | | | |
| **Assess personal context** | • Learn about the pregnancy and assess current readiness to quit (S1)<br>• Assess current and past smoking (S1)<br>• Assess physiological and mental functioning (S1)<br>• Establish personal reasons for quitting and confidence in ability to quit (S1)<br>• Assess past quit attempts (S1)<br>• Establish understanding of how smoking affects pregnancy (S1) | EASY (7.1) | • Pre-quit questionnaire<br>• Core questions plus optional questions for enhanced tailoring | • Consistency<br>• Potential to improve accuracy of self-reporting due to anonymity | • Can lack nuance<br>• Difficult to capture broader social and environmental influences on smoking behaviour<br>• Too many written questions might discourage engagement |
| **Relationship and rapport** | • Build rapport (S1-S6)<br>• Reflective listening and respond appropriately (S1-S6)<br>•Provide reassurance (S1-S6) | HARD (3.3) | • Tailoring and personalisation<br>• Peer support<br>• Chatbot<br>• Human advisor component | • Support without human interaction<br>• Good rapport is not universal in interpersonal support<br>• Non-judgemental | • Lack of human accountability<br>• Lack of human empathy<br>• More difficult to adapt to context |
| **Provide information** | • Discuss the importance of quitting with support and inform about the treatment programme (S1)<br>• Explain how tobacco dependence develops and assess nicotine addiction (S1)<br>• Explain the importance of abrupt cessation and the 'not a puff' rule (S1)<br>• Inform about withdrawal symptoms (S1)<br>• Discuss potential withdrawal symptoms and cravings / urges to smoke and how to deal with them (S2) | EASY (8.8) | • Text<br>• Video<br>• Images / infographics<br>• Audio | • Combination of written, verbal and visual information<br>• Consistency<br>• Convenience<br>• Able to revisit and share information with others | • Level of digital literacy<br>• Engagement |
| **Check understanding and answer questions** | • Check information and instructions have been understood (S1-S6) | MEDIUM (4.7) | <u>Check understanding</u><br>• Quiz<br>• Reflexive questions<br>• Interactive video<br><u>Answer questions</u><br>• Frequently asked questions<br>• Chatbot (with ability to escalate to a human) | • Consistency<br>• Creative methods could help to embed learning | • Could feel like a test/ be discouraging if get answers wrong<br>• Misunderstanding might be harder to identify<br>• Not always clear what questions need to be asked<br>• Engagement |

*(Continued)*

**Table 2.** (*Continued*)

| Plan quit | • Set a quit date (S1)<br>• Prompt and elicit commitment (S1&S2)<br>• Confirm readiness and ability to quit (S1&S2)<br>• Advise on changing routines (S2)<br>• Discuss who will be able to offer support during quit attempt (S2)<br>• Discuss smoking contacts and available supports (S1&S2)<br>• Address any potential high-risk situations in the coming week (S2)<br>• Confirm the importance of abrupt cessation (S2)<br>• Discuss preparations/plans and provide summary (S1, S2)<br>• Confirm the importance of the 'not a puff' rule and prompt commitment (S3-6) | MEDIUM (6.7) | • Quit plan questionnaire<br>• Dynamic goal setting questionnaire | • Visual quit plan<br>• Visual list of goals<br>• Schedule notifications | • Adaptive programming increases development complexity (e.g. not having to set a quit date at the outset, accommodating quit date changes and restarts with different content) |
|---|---|---|---|---|---|
| Check progress | • Check on how pregnancy is progressing (S2)<br>• Check on progress (e.g. smoking status) (S3-S6)<br>• Discuss any experienced withdrawal symptoms and cravings / urges to smoke (S3-S6)<br>• Discuss any difficult situations experienced and methods of coping (S3-S6) | MEDIUM (6.4) | • Ongoing monitoring data<br>• Questionnaires<br>• Smiley face feedback<br>• A chatbot | • Self-monitoring can be a motivational tool | • Too many written questions might discourage engagement<br>• Difficult to capture broader social and environmental influences on smoking behaviour |
| Address progress | • Address how to deal with withdrawal symptoms and cravings / urges to smoke (S2-S6)<br>• Address any potential high-risk situations in the coming week/future (S3-S6)<br>• Provide summary (S3-S6) | MEDIUM (5.9) | • Tailored information and advice in response to ongoing monitoring data and individual preferences | • Tailored to the individual | • More difficult to manage people who are struggling to quit/ have relapsed<br>• Tailoring is dependent on the level of self-reporting |
| Future support | • Assess risk of relapse, provide motivation and support (S6)<br>• Assess woman's individual needs for ongoing and agree to plan for follow-up support and/or next appointment (S6) | MEDIUM (5.8) | • Quit support final summary focusing on relapse prevention<br>• Intermittent check ins<br>• Restart support or signpost to other support options | • Visual personal summary<br>• Prompts for re-engagement | • Adaptive programming increases development complexity (e.g. providing different content to people who relapse)<br>• Need for local knowledge e.g. referral to services if app unsuccessful |
| **BIOCHEMICAL FEEDBACK** | | | | | |
| **Carbon monoxide monitoring** | • Explain and conduct CO monitoring (S1)<br>• Conduct CO monitoring (S2-S6) | HARD* (4.0) (* not all participants were aware of personal CO monitors) | • Remote ICO monitors<br>• Potential to link with midwife-led smokefree pregnancy pathway | • Self-monitoring can be a motivational tool<br>• Ownership of CO readings<br>• Accessibility | • Cost for the technology<br>• Technical issues<br>• Digital literacy |

(*Continued*)

**Table 2.** (Continued)

| NICOTINE SUBSTITUTION. PRODUCTS | | | | |
|---|---|---|---|---|
| **NRT & e-cigarettes information and demonstrate use** | • Discuss the use of nicotine replacement therapy and e-cigarette (S1) | EASY (7.1) | • Product decision support tool<br>• Product demonstration videos | • Timely and can revisit at any time | • More challenging to check technique |
| **NRT & e-cigarette supply** | • Confirm availability of NRT or e-cigarette supplies and discuss expectations of products (S2)<br>• Enquire about NRT or e-cigarette use and ensure sufficient supply (S3)<br>• Advise about continued NRT or e-cigarette use and where to obtain further supplies (S6) | MEDIUM (6.5) | • Electronic ordering system<br>• NRT delivered by post or local pharmacy collection | • Accessibility | • Governance implications for supplying nicotine substitution products without health professional supervision<br>• Cannot set an immediate quite date (postal)<br>• Quantity limits needed to minimise waste (postal)<br>• People may misrepresent eligibility if there are no identity check (postal) |

(S1-S6 = Session 1 - Session 6)

word 'conversation'. It's face to face, telephone, even on a text messaging you can have a conversation because you're interactive. On an app can you get that level of interaction?"-FG3-08-PTC).

**Information provision.** Providing information was considered the easiest aspect of the STP to deliver digitally, with several potential advantages identified. Firstly, participants suggested it could improve consistency, ensuring that everyone has access to the same evidence-based information ("*The advantage in there is you can make it the absolute best advisor.*"-FG5-14-PTC). It might also improve flexibility, so that people can explore information in their own time, revisit it or share with others, including partners and significant people in their lives. Participants suggested that a key advantage of digital is "*there's something for everyone*" (FG3-08-PTC). Information can be delivered in different ways to suit individual preferences, such as text, images and videos ("*Everybody responds different, some people are more kind of auditory or visual, so I think it offers flexibility in how that information can be given.*"-FG3-07-PTC). Despite these advantages, there was said to be no guarantee people would engage with the information provided or use it appropriately.

## Planning the quit attempt (excluding nicotine substitution products)

A digital intervention was seen to have several advantages over interpersonal support when it came to planning a quit attempt. Participants suggested that it could be tailored to individual needs, for example, "*...scheduling particular notifications, recording yourself little videos, having pre-commitment devices, I think that sort of thing can be done digitally quite nicely...*" (IV2-06-AC). Others talked about various self-assessments that could be incorporated, such as goal setting, identifying smoking triggers, and confirming readiness to quit. Digital tools might also help to summarise the quit plan and "*make it very visual*" (FG9-28-PR).

While choosing a quit date was felt to be relatively straightforward ("*It'll be fairly easy to have a diary and people choose their quit date...*"-FG1-02-AC), many wondered how a digital intervention would support pregnant smokers on the spectrum of quitting. The urgency around quitting smoking in pregnancy was apparent, but participants said that pregnant smokers are not always ready to set a quit date ("*...so what are the options for people who don't want to quit but do want to sort of do something? [...] a lot of times women don't want to set a date they just want to reduce.*"-FG2-05-AC). As such, requiring a quit date to initiate the digital

intervention could be a trigger for disengagement ("*. . .if you did try to make them commit to a quit date on an app they might just go 'Oh I don't know how I feel about this. I'm going to have a little think about it, I'll come back to it' and they would potentially just then not log back onto it again because it's asking them to commit. . .*"-FG9-28-PR). However, on the other hand, some participants questioned whether there would be a time limit on how far in the future a quit date could be set ("*I also see the risk there that if someone chooses a quit day two months later and that's accepted.*"-FG9-27-PR).

Digital support was considered to have some limitations when it came to planning the quit as an automated system can only manage what has been programmed ("*. . .you can't anticipate what everybody is thinking all the time.*"-FG2-05-AC). When a quit attempt does not go to plan, practitioners said they would explore what went wrong and were unclear how this could be replicated digitally ("*I'm thinking about the quit date and that quit date comes and the stopping smoking doesn't happen and those conversations around that time are really, I don't know, how you could do that digitally?*"-FG9-29-PR). Indeed, it was argued that most digital support to date has been built on the assumption that an individual will set a quit date and quit, whereas people often relapse or do not want to fully quit ("*. . .people are not robots so they might quit and then go back and then relapse and then want to go again and all of that flexibility is really important but very hard sometimes to do in an app.*"-FG2-05-AC).

**Checking understanding.**    Various ideas to digitally check for understanding were suggested. Simply asking pregnant smokers if they understood the information was seen as insufficient as this would not identify if it had been misunderstood. Quizzes and asking reflective questions were seen as the main ways to ensure that learning was happening. However, participants cautioned that this might be discouraging if it is seen as a test ("*. . .you've failed this, two out of four, redo it.*"-FG8-22-PR). Getting engagement right with "*presentation nuance*" (FG8-26-PTC) and ensuring that gaps in knowledge are addressed in a supportive way were both seen as critical. Depending on the resources available for intervention development, some participants suggested more interactive approaches, such as games, picture drag and drop activities, or avatar based 'choose the story' videos ("*. . .basically putting yourself in someone else's shoes and taking them through that journey, so there's a bit of a dissociation but you're also testing that they've understood. . .*"-FG5-13-PTC). However, others saw understanding as more dynamic than comprehension alone, suggesting that it involves getting people to reflect and make use of the learning, and this might be harder to do digitally.

Answering pregnant smokers' questions using digital tools was considered more difficult. On a basic level, participants said that a 'frequently asked questions' page could be created but this should not remain static. Others thought this might be an opportunity to use a chatbot to simulate a human-like conversation. However, while this might work for basic issues, some practitioners argued that there is often a lot of "*reading between the lines*" (FG11-34-PR) as pregnant smokers do not always know what questions they need to ask. Others said that while chatbots may help to provide initial triage, they can create a negative experience if the chatbot does not understand the request and therefore, a form of human interaction for escalations might be needed ("*. . .it's complementing a follow-up with a person like, a call or a live chat function. . .*"-FG2-06-AC). Engagement with a chatbot may also depend on interaction style preference and perceived support needs.

**Checking progress.**    Participants were equivocal about how easy it would be to check progress digitally. On the one hand, various methods were suggested to establish how the quit attempt was going. These included standard questionnaires, smiley face feedback, a chatbot, CO readings and daily monitoring of tobacco and nicotine use. The latter two of these would also allow pregnant smokers to visually track their own progress, which was considered a particular advantage of a digital intervention. However, because of the multitude of reasons why

someone might be struggling to quit, "*You've got your smoking, your NRT use, your withdrawal, your cravings. . .*" (FG3-07-PTC) this meant that often *"there's a lot to unpack"* (FG3-07-PTC). To capture this digitally would involve effort on the part of the pregnant person to enter relevant information ("*. . .a lot of finger pressing and sort of it won't necessarily feel very natural like a conversation does.*"-IV2-37-AC). Like with interpersonal support, a digital system can only work with the accuracy and completeness of information provided. Relying on pregnant smokers to regularly answer digital questions was seen as the biggest challenge to translating this feature of the STP, and even then, it may not be possible to capture the level of detail and context that a human discussion allows ("*. . .how can you discuss something if you're putting, if it's a yes or no question, if it's closed. . .*"-FG6-16-PR). Consequently, some participants suggested that there might need to be a "*do you need to speak to a specialist*?" (FG5-14-PTC) feature built in.

Prompts and reminders were identified as a good way to encourage people to track progress and sustain engagement ("*. . .they get a little nudge [. . .] it pings up on the phone*"-FG7-20-PR). However, it was expected that people would disengage as with interpersonal cessation support ("*Quite how well that [a nudge] would land with pregnant women who were struggling [. . .] they've ducked out of it because they're not motivated to stay in. . .*"-FG3-08-PTC).

**Addressing progress.** Participants said that tracking data and other information about progress could be used to tailor information and advice *(". . .having tailored responses on how people respond, I think that probably works quite well. . ."*-FG2-06-AC). It may also be possible to track whether people have looked at follow-up support and send them reminders if not ("*. . .this is a really useful thing to go and look at and to know if they've looked at it.*"-FG7-18-AC). Participants felt that the main limitations of addressing progress digitally were that it would be restricted to smoking, whereas stop smoking advisors often deal with wider issues that may impact on smoking behaviour, and managing people who have failed to stop or relapsed ("*It might be quite difficult to address that in a meaningful way and it not just to feel like you know it's the same thing again.*"-FG2-06-AC).

**Future support.** In the STP guidance, the 4-week post quit date session is described as an opportunity to identify a person's individual needs for ongoing support and plan for follow-up support. In practice, the offer of ongoing support was said to vary according to local protocols and procedures, along with how well the quit is going. There was some debate over the optimum length for a digital cessation intervention and that this may depend on the features included ("*. . .is it forever? Is there a length of time that they are engaged or is it just something that could be an ongoing community of support?*"-FG4-09-PTC). Participants indicated that digital support lends itself to creating a personalised "*discharge summary*" (FG3-08-PTC). If the quit has gone well, then this could focus on relapse prevention along with an occasional check in to restart support if needed, but if people had struggled to quit, then restarting digital support may not be appropriate and information on alternative support options could be provided. Some participants suggested it may be easier for those who have relapsed to seek out support again anonymously through a digital intervention. Others disagreed and thought that rapport with a stop smoking advisor is important for restarting a quit ("*. . .that whole relationship that they've built with their practitioner during the time that they've been having support is why they call their practitioner at that point in time and say 'I've had a blip' [. . .] and you wouldn't have that with an app*."-FG11-35-PR).

## Theme 2: Remote digital provision of CO monitoring and nicotine substitution products

**Carbon Monoxide (CO) monitoring.** The extent to which CO monitoring was seen as a necessary part of a digital cessation intervention in pregnancy depended on whether this was

being routinely offered in antenatal checks ("*. . .it would be a shame not to have it but equally I can see that maybe you don't necessarily need it [. . .] you'll still be seeing your midwife if that's something that the midwife could do and you've got digital support outside of that, then maybe it's not as important.*"-FG1-01-AC). Smartphone-linked personal CO monitors were suggested as a potential alternative to allow pregnant smokers to independently monitor and track CO levels without seeing a healthcare professional. There was substantial variation in participants' awareness of personal CO monitors. Those who were familiar with them thought they would be relatively easy to integrate into a digital cessation intervention and could enhance the offer ("*I think the iCO [personal CO monitor], something that's an interactive tool alongside digital you know it's another element isn't it [. . .] it brings another dimension to digital*"-FG5-14-PTR). However, practitioners who had used personal CO monitors with clients described them as "*hit and miss*" (FG10-32-PR); some reported positive experiences, believing they helped to increase motivation to quit, but others experienced low uptake and usage issues ("*. . .some of them are raving about it and think it's brilliant and others are just not really interested in it or find it a bit clunky to use*"-FG10-32-PR).

It was suggested that the personal CO monitors might give pregnant smokers ownership of readings, an ability to track changing levels, and the immediacy of feedback might make the impacts more salient ("*So if you give them something that they can use privately and they can see 'Oh actually when I have a cigarette my carbon monoxide is really high and it does go down like she said it would' [. . .] It makes it very palpably real. . .*"-FG9-28-PR). The main reservation against personal CO monitors were the resources involved, especially as few services could currently afford them. However, others argued that the initial outlay was minimal compared to the potential long-term savings ("*. . .I think people go 'Oh God is it fifty or sixty pounds for one?' Well actually if that woman had a high-risk birth it would cost us ten and a half thousand compared to three and a half thousand if she was a low-risk birth. . .*"-FG6-16-PR). Concerns were also expressed about the use of personal CO monitors to validate quits remotely, especially when associated with incentive schemes, as the devices could be misused.

Finally, some participants discussed the possibility of results being shared with the healthcare professionals supporting the pregnancy and linking with maternity records, and that this could enhance the tailoring of support and provide accountability–as part of a "*dynamic relationship*" (FG4-09-PTC). However, achieving such integration would be reliant on services responding in a timely manner to a high reading or inactivity and despite support for the idea in theory, participants noted that current systems did not easily enable this. In contrast, some thought that this might limit use of personal CO monitoring because it would feel like the "*CO monitoring is checking up on them rather than a motivational tool*" (FG4-12-PR), "*making people feel that big brother is watching.*" (FG4-09-PTC).

**Views on remote provision of nicotine substitution products without an advisor.** Most experts were unconcerned about providing NRT remotely without an advisor and believed it would enhance a digital cessation intervention. Additionally, it was thought that the provision of e-cigarettes should be considered given their increasing use for smoking cessation among pregnant smokers in the UK. It was argued that both NRT and e-cigarettes can easily be purchased without a prescription, and that making them available online to order might improve timely access ("*I think you can make NRT supply really easy through digital and I think you can make the vape offer really easy through digital, so I don't see that as a barrier [. . .] You can buy them at the shops, can't you?*"-FG5-14-PTC,). However, some participants had reservations, particularly those involved in commissioning. One public health expert described it as a "*step too far*" (FG4-09-PTC), highlighting the potential governance and regulatory issues involved for those supplying the nicotine substitution products ("*. . .what happens if it then goes wrong and how to make sure that they're getting the right amount and the right product in the right*

*way at the right time. . ."*-FG4-09-PTC). In addition, safety concerns were expressed by some participants from countries where NRT and e-cigarette use in pregnancy are not supported.

Several practitioners were unclear how nicotine substitution products could be supplied without a prior consultation with a qualified professional, much in the same way as prescribed medications; though they could see some benefits to pregnant smokers being able to subsequently order products electronically ("*. . .you can go on [NHS app] and renew prescriptions and things like that so I've just imagined something similar. . ."*-FG1-02-AC).

**NRT and e-cigarette product choice and fulfilment approaches.** NRT product choice was said to be nuanced and informed by personal beliefs and experience, which practitioners would typically elicit and address though conversation ("*With choosing NRT products, again I think that's quite tricky to do via an app because chances are they've got their own experiences of NRT products before now and quite often say 'No' to some products because of previous experience and you quite often need to explore what that previous experience was. . ."*-FG9-27-PR). Likewise, there are many different types of e-cigarettes which might make supporting product choice more challenging.

It was generally agreed that it would be easy to demonstrate nicotine substitution products digitally, with the most popular method being videos. This method is often used for other medicinal products and some practitioners already signposted to NRT videos or directed people to e-cigarette manufacturer websites. A perceived advantage of this method was that it is on demand and people can replay it at any time. Another suggested advantage of videos is that they can easily be translated into different languages, and the visual medium can convey the information more easily than words alone, enhancing inclusivity (*. . .that's really what we found you know it doesn't matter if there's a language barrier or anything else you know it's just really simple and easy."*-FG10-32-PR). In contrast, a few participants believed this would be less effective than in-person as you cannot check whether someone is using the products correctly ("*I mean despite getting the information, despite seeing the video, actually a lot of people still use it incorrectly."*-FG9-27-PR).

Two main fulfilment approaches were suggested. Firstly, posting NRT, and potentially e-cigarettes, directly to pregnant smokers; secondly, sending a request to a nominated pharmacy for collection. Both approaches were already used by services, and advantages and disadvantages were identified for each. Posting products was said to remove more barriers to access and some services suggested it had increased the number of successful quit attempts ("*. . .when I first started out, we sent people to the pharmacy to collect their NRT [. . .] then we went to direct supply and we saw a huge increase in quitting. . ."*-FG11-35-PR). However, this approach was more costly, especially as pregnant smokers tend to switch products ("*. . .they want to change products so it gets quite expensive if you're issuing a lot and then two days in it's, 'No, I can't use that'."*-FG7-18-AC). One service that had commissioned NRT provision alongside a generic cessation app said they were charged a handling fee for each order ("*. . .if you've got a £15 handling fee for example every single time someone is getting dispensed something new, it's going to get quite expensive."*-FG10-32-PR). However, this did overcome issues with commissioning and having the right infrastructure in place to manage and distribute orders. It was also noted that people cannot set an immediate quit date with postal supply ("*. . .they're ready to stop and we have to explain to them that I can't magic it to you."*-FG7-18-AC) and it would need "*. . .somebody with the right qualification to be signed off."* (FG7-19-PR). Linking with pharmacies for NRT provision would make the sign off process simpler and ensure that there was someone to discuss the products with and support use. However, this approach would require pharmacies to be signed up to the scheme and would involve travel making it potentially less accessible.

Finally, some participants were concerned that without an identity verification process, people might try to misrepresent their eligibility to obtain free nicotine substitution products (*"We've had this yesterday with our app* [non-pregnancy] *that somebody has tried signing on with three different names to try and get vaping and e-liquids and NRT [. . .] there's a few risks around people just being able to order and not have some governance around it. . .it does cause me some anxiety."*-FG6-16-PR).

**Potential impact on NRT adherence.** A key discussion point was the potential for remote NRT supply to have a negative impact on NRT adherence (*". . .we know that [adherence] is already a problem and if people are nervous about using the products anyway before they even sort of look at them, then I'd see that as an even bigger barrier if there isn't that interpersonal explanation, reassurance, helping people to use the products correctly. . ."*-FG8-26-PTC). A key part of follow-up was said to be exploring how people are getting on with their products and troubleshooting problems. If these are not addressed in a reactive way then pregnant smokers may discontinue use (*"It can take a good switching around of products and regular contact to kind of get that right for someone [. . .] that can really make or break whether someone carries on engaging, so I think it would need to be quite reactive in the way it worked. . ."*-FG10-34-PR). Additionally, it was suggested that it may be more challenging to encourage use of combination NRT if pregnant smokers are deciding this for themselves (*". . .for people that don't necessarily want more than one, you'll like try and encourage them to try another one as well."*-FG7-18-AC).

## Theme 3: Integration and reach

**Integration with standard care.** Participants discussed the extent to which a digital intervention might reinforce interpersonal stop smoking support as *"part of the kit bag"* (FG5-14-PTC) or be an alternative by *"filling in gaps"* (FG4-10-PTC). Most practitioners indicated that digital support could have an impactful role as an adjunct to existing interpersonal support (*". . .having electronic support alongside an advisor is probably like the strongest combination you could have. . ."*-FG9-28-PR). For example, it could reinforce information, provide additional quit tools or even integrate data with local maternity services. It was felt this holistic systems approach is often overlooked when developing digital interventions (*"If there was something that was very nicely integrated with treatment, I think that's missing. . . whereas everything is sort of like you can also use it, but it has nothing to do with what you're doing with your counsellors."*-FG2-05-AC).

Others felt that a standalone intervention could be advantageous as it allows a sense of ownership and potentially provides a private, less judgemental space (*"I think you always need to have that existing provision but there are always going to be women who don't want to access that and so it's quite important to be able to provide something for those women as well."*-FG2-04-AC).

Some practitioners cautioned that a digital cessation intervention should not be seen as a low-cost replacement of the existing maternity services stop smoking pathway (*". . .locally, you'd get some commissioners who think 'I can save some money here by decommissioning those face-to-face services and just put some money into the app'"*-FG2-04-AC). Moreover, it was felt that a digital intervention could inadvertently contribute towards disengagement with services because pregnant smokers see it as an opportunity to 'opt out' of interpersonal support (*"I think this is a fabulous idea but I don't want people to think 'Oh, it's alright I don't have to engage with maternity services because I've got this app' [. . .] I don't want it [interpersonal support] to [. . .] become the alternative rather than the default model of support for anybody who's pregnant. . ."*-FG4-09-PTC).

**Reach.** Participants discussed the potential of a digital intervention to extend reach and engage with different populations. It could offer flexibility, out-of-hours support, and overcome geographic and practical barriers to accessing services. However, a predominant view was that it is likely to be those who are already motivated and would engage with services anyway who would use a digital intervention ("*. . .trying to engage women who are more ambivalent and/ or who don't want to stop will be extremely difficult.*"- IV2-37-AV*). Reaching and engaging partners and significant others who smoke to encourage them to quit was also identified as a gap which a digital intervention could potentially help fill ("*. . .everything is focused on the women where a lot of the barrier is around her support system, and if there's any way to engage in the app to have material that is directed at the partner or other people that she's living with, that could be very helpful.*"-FG2-05-AC).

Clear themes were evident around the potential for digital support to fill a timing gap in pregnancy cessation support, specifically around preconception and early pregnancy ("*Well obviously the sooner the better isn't it really? But then they are supposed to be offered it at their booking [first appointment with midwife], but I guess women may be searching for support even before that. . .*"-FG2-04-AC). Supporting those who spontaneously quit in early pregnancy but may be susceptible to relapse, and sustaining longer term support and relapse prevention postpartum, were also seen as areas where a digital intervention could help reinforce the importance of continued abstinence.

## Discussion

This study explored experts' views on the potential transferability of UK STP for smoking cessation in pregnancy into a comprehensive digital cessation intervention. Experts were supportive of the idea and felt that most behavioural counselling content from the STP could be automated. Digital support was perceived to have several advantages over interpersonal support when it came to the needs of pregnant smokers, such as increasing accessibility and consistency, removing social stigma, and empowering users to manage their own quit journey through tracking and tailoring. These findings build on and extend previous qualitative research with health professionals which identified several potential benefits to digital cessation interventions in pregnancy (e.g. providing anonymity and being available on demand) [25].

However, experts expressed concerns that the absence of relationship factors, such as interpersonal accountability to a human advisor and empathy for the situational challenges surrounding a quit attempt, could undermine engagement with and effective use of a digital intervention. It was suggested that there may need to be some level of human involvement. This view is consistent with the 'Supportive Accountability' model [41] which argues that adherence to digital interventions is increased by human support. The model draws on self-determination theory, proposing that the need for human support is moderated by motivational factors; those who are more intrinsically motivated to quit will less likely require human support to adhere to a digital intervention. However, there has been very little testing of this model on digital cessation interventions for pregnancy. One observational study looking at maternal smoking in low-income communities found a positive relationship between counsellor advice, accessed via a weblink-portal in an app, and app usage [42], but the non-experimental design limits generalisability. Additionally, it must be recognised that the inclusion of human support increases implementation costs and thereby reduces scalability. For this reason, it is important to consider when and how a digital cessation intervention needs to be supplemented by human support or if alternative lines of accountability, rapport and adaptive responding can be established.

Artificial intelligence (AI) chatbots might offer a solution to replicating human-like interactions. There is some promising evidence that chatbots increase user engagement with generic digital cessation interventions [43,44] and help foster a sense of accountability [45]. However, among the existing studies there is variation in the sophistication of chatbot technologies and how they are delivered which might have implications not only for engagement and effectiveness with them as a cessation tool, but for development and maintenance costs. Likewise, the more complex the overall intervention becomes, and the more 'active' engagement from users, the greater the requirement is for digital literacy skills and competencies.

Peer support was also among the experts' suggestions as an approach to substitute the relational aspects of advisor-delivered support. A peer support relationship is based on the idea of *reciprocal* accountability where people help to motivate and learn from one another. Qualitative studies have shown that pregnant smokers have a strong preference for contact with others in the same situation as themselves [46,47]. While it has been theorised that peer support might enhance engagement with digital interventions [48], evidence for the effectiveness of peer support as a cessation intervention for pregnancy is uncertain [9].

Other suggested technology-driven strategies to help foster a digital therapeutic relationship included tailoring (creating individual strategies and feedback for a person) and personalisation (taking information about a person or sender and adding it to content e.g. name). Several studies have found that tailored and personalised digital health interventions are more effective for behaviour change [49,50], particularly if the tailoring and personalisation occur dynamically throughout the intervention as this makes it more relevant and meaningful [51]. Tailored internet and text-based digital cessation interventions targeted to pregnancy have shown evidence of cessation efficacy [29]. However, more recently published evidence from the 'MiQuit' trial suggests that while tailored text-messaging interventions increase attempts to quit smoking in pregnancy, they may not be sufficient for maintaining a quit [52]. Indeed, some of the limitations of digital support that were identified by experts, such as not adapting content after an unsuccessful quit, may be more of a reflection on current interventions rather than the possibilities of digital support. A comprehensive digital intervention that brings together multiple evidence-based components and dynamically tailors content could potentially overcome these issues.

Smartphone-linked personal CO monitors are now available which provide real-time monitoring data to users. These could be used to replicate the CO monitoring component in the STP. Experts were conflicted about the need to include remote CO monitoring in a digital intervention. It could be an attractive feature for pregnant smokers and an additional tool to create engagement, but this was balanced against the potential cost and technical issues. Another view was that in-person CO monitoring should already be routinely offered as part of the Saving Babies Lives Care Bundles(v3) [53]. Examples of generic digital cessation interventions which have integrated personal CO monitors to boost motivation exist [54] and a small-scale UK stop smoking service evaluation indicates that personal CO monitors might support 'self-care skills' for cessation in pregnancy [55]. However, higher quality evidence is needed as the cost-effectiveness of these devices as a motivational tool remains unclear [56,57].

Remote provision of NRT, and potentially vapes, was generally well supported. However, key concerns were voiced around governance issues for those providing nicotine products to pregnant smokers without a consultation with a trained advisor (e.g. ensuring it is provided in line with NICE guidelines and local protocols). There has been much learning around the remote provision of NRT during the COVID pandemic and outside of pregnancy there is evidence that suggests smokers prefer quit aids to be directly posted to them [58]. What is not known, however, is how effective a decision support tool for nicotine product choice might be. A further concern was about the potential impact on NRT adherence. However, there may be

opportunities for innovation in utilising digital approaches to improve NRT adherence in pregnancy. One such intervention, Baby, Me & NRT, which includes a digital text-message component, is currently being tested in a randomised control trial [59]. Finally, experts cautioned that people might try to cheat the system to obtain free nicotine products. Misrepresentation is a growing issue for digital health interventions where some people may use deception to get access to technologies, medications or incentives, especially when these are advertised online [60]. Precautions are therefore needed to verify identities and the personal information provided.

There was much discussion about whether the digital intervention should be a standalone tool or integrated with the local maternity systems smokefree pregnancy pathway. On the one hand, a standalone tool may help to overcome issues of autonomy and social stigma, but on the other, an integrated tool with health provider dashboard could enable data-informed conversations about smoking between health professionals and pregnant smokers. However, developing a digital intervention that would work seamlessly alongside various NHS data systems, or getting services to adopt a new system, was recognised as a major implementation challenge.

Finally, it was noted that partners and significant others often have an important but overlooked role in pregnancy quit attempts. Current UK clinical guidance [12] includes recommendations for providers of cessation services to support partners but, to our knowledge, this has not been incorporated into digital cessation interventions for pregnancy so should be assessed in future research.

### Strengths and limitations

Study strengths include the sample size and diverse range of experts involved. The small group format also enabled a more comprehensive and rich exploration of a range of issues related to translating the STP into a comprehensive digital intervention. While we attempted to achieve maximum variation in participant characteristics in terms of expert group and geography, our recruitment approach also had disadvantages. The sample was not chosen through random selection and therefore it may not be representative of the target population, which limits the generalisability of the findings. The analytical approach is subject to interpretation, although we used double coding on a sample of transcripts and discussed the findings as a research team to ensure that interpretations remained grounded in the data. Additionally, consideration must be given to the limitations of digitalising the STP for smoking in pregnancy. Addressing the social determinants of smoking requires a broader range of actions than an individual-level intervention can address alone. Additional population-level interventions will therefore continue to be needed [61]. However, a digital intervention offers the potential to enable greater equity in access to stop smoking services in pregnancy. Strategies can also be embedded within the digital intervention to help respond to social context such as tailoring to individual characteristics, providing coping strategies, offering support for family and friends, and signposting to other support services.

### Conclusion

The STP was considered largely transferable to a digital intervention and would potentially be helpful to pregnant smokers alongside, or as an alternative to, standard care. There were concerns that the relational aspect of advisor-led support, such as accountability, rapport, and adaptive responding to context, would be hard to replicate digitally, though various opportunities exist to make digital interventions more human-like. Likewise, CO monitoring and the supply of NRT and potentially e-cigarettes (if these become standard care in the future based

on safety and effectiveness evidence) could be delivered remotely but would require careful consideration of the potential pitfalls, such as technical issues and the ability to misrepresent eligibility. A key theme was the need to balance the complexities of developing and implementing an optimum digital intervention with cost-effectiveness. Better understanding of what factors inhibit or enable engagement also appears essential. The findings suggest that a comprehensive digital intervention for smoking cessation in pregnancy merits further development and evaluation.

## Supporting information

**S1 File. Example Mentimeter activity.**
(DOCX)

**S2 File. Focus group topic guide for experts.**
(DOCX)

## Acknowledgments

We would like to thank the experts who generously gave their time to take part in this study, our public involvement advisory panel and our programme advisory group (Sophia Papadakis, Jo Locker and Julia Robson) for providing insights and guidance throughout. The views expressed are those of the authors and not necessarily those of the NIHR or the Department of Health and Social Care.

## Author Contributions

**Conceptualization:** Lisa McDaid, Joanne Emery, Tim Coleman, Jo Leonardi-Bee, Felix Naughton.

**Formal analysis:** Lisa McDaid, Pippa Belderson.

**Funding acquisition:** Lisa McDaid, Joanne Emery, Tim Coleman, Jo Leonardi-Bee, Felix Naughton.

**Investigation:** Lisa McDaid, Pippa Belderson.

**Methodology:** Lisa McDaid, Pippa Belderson, Joanne Emery, Tim Coleman, Felix Naughton.

**Project administration:** Lisa McDaid, Pippa Belderson.

**Supervision:** Tim Coleman, Felix Naughton.

**Writing – original draft:** Lisa McDaid.

**Writing – review & editing:** Lisa McDaid, Pippa Belderson, Joanne Emery, Tim Coleman, Jo Leonardi-Bee, Felix Naughton.

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
