## [Decision Letter · Decision Letter 0]

10 Aug 2023

PDIG-D-23-00250

Experts’ views on translating NHS support to stop smoking in pregnancy into a comprehensive digital intervention

PLOS Digital Health

Dear Dr. McDaid,

Thank you for submitting your manuscript to PLOS Digital Health. After careful consideration, we feel that it has merit but does not fully meet PLOS Digital Health's publication criteria as it currently stands. Therefore, we invite you to submit a revised version of the manuscript that addresses the points raised during the review process.

Please submit your revised manuscript within 60 days Oct 09 2023 11:59PM. If you will need more time than this to complete your revisions, please reply to this message or contact the journal office at digitalhealth@plos.org. Please include the following items when submitting your revised manuscript:

We look forward to receiving your revised manuscript.

Kind regards,

Haleh Ayatollahi

Section Editor

PLOS Digital Health

Journal Requirements:

b. If any authors received a salary from any of your funders, please state which authors and which funders.

2. We have noticed that you have uploaded Supporting Information files, but you have not included a list of legends. Please add a full list of legends for your Supporting Information files after the references list.

Additional Editor Comments (if provided):

The manuscript was interesting. Please address the following comments in your revision.

1- Please add appropriate keywords using MeSH terms.

2- Please follow the journal instructions for using appropriate headings, subheadings and organizing the manuscript.

3- The manuscript is a bit long. Please summarize the content or remove unnecessary paragraphs.

Reviewers' comments:

Reviewer's Responses to Questions

**Comments to the Author**

1. Does this manuscript meet PLOS Digital Health’s publication criteria? Is the manuscript technically sound, and do the data support the conclusions? The manuscript must describe methodologically and ethically rigorous research with conclusions that are appropriately drawn based on the data presented.

Reviewer #1: Yes

Reviewer #2: Yes

Reviewer #3: Partly

2. Has the statistical analysis been performed appropriately and rigorously?

Reviewer #1: Yes

Reviewer #2: N/A

Reviewer #3: N/A

3. Have the authors made all data underlying the findings in their manuscript fully available (please refer to the Data Availability Statement at the start of the manuscript PDF file)?

Reviewer #1: Yes

Reviewer #2: Yes

Reviewer #3: No

4. Is the manuscript presented in an intelligible fashion and written in standard English?

Reviewer #1: Yes

Reviewer #2: Yes

Reviewer #3: Yes

5. Review Comments to the Author

Reviewer #1: This is a very interesting approach to the analysis and implementation of what seems a very useful tool. The only concern is if it would be actually implemented, would the patients lose personalized care to some extent? It is important not to lose contact with pregnant women who as we know can present serious complications specially during the third trimester. 

Thank you for your submission.

Reviewer #2: Page no-13 and may other areas: Author suggested e-cigarettes as a substitute of nicotine but not mentioned any health risk and safety issue of e-cigarettes during pregnancy as most e-cigarettes contain nicotine and other harmful substances besides nicotine. (1,2)

References:

1. Quick facts on the risks of e-cigarettes for kids, teens, and young adults [Internet]. Centers for Disease Control and Prevention; 2022 [cited 2023 Aug 8]. Available from: https://www.cdc.gov/tobacco/basic_information/e-cigarettes/Quick-Facts-on-the-Risks-of-E-cigarettes-for-Kids-Teens-and-Young-Adults.html

2. Association AL. Health risks of e-cigarettes and vaping [Internet]. [cited 2023 Aug 8]. Available from: https://www.lung.org/quit-smoking/e-cigarettes-vaping/impact-of-e-cigarettes-on-lung

Note>1: Most of the participants of the of study argued about the future application of the proposed comprehensive digital intervention due to digital literacy, cost, technical issues, manipulation of the system by the users etc. So these should be considered and mentioned in the conclusion section. 

Note>2: Involvement of Biomedical engineering experts to the procedure (both for study and review process) is recommended for identifying the health risk as well as technical issues

Reviewer #3: Cessation of smoking tobacco during pregnancy is one key preventive approach for positive obstetrical outcome. The intent of the study is commendable. However, my observations and concerns are as follows: 

1. It is a hypothetical study to me with several non-corroborative subjective methods

2. How digital health can address this issue is largely unclear to me. Authors are requested to cite examples

3. Duration of smoking and number of cigarettes smoked/day are two crucial criteria for assistive cessation of smoking habits. In the article, I haven't seen the mention of these while selecting the participants. Age group, comorbidities, and social determinants of health parameters are not included too, which I believe could influence the outcome very much. 

The neurobiology of smoking habit (although in a different perspective) can be cited "Chattopadhyay, S. (2022). Cigarette Smoking Erectile Dysfunction and its Extended Psychobiology. Adv Sex Reprod Health Res, 1(1), 124 -127." 

4. How 'experts' are defined in this work is unclear. What are their experience levels in years? and most importantly, what are the anomalies and commonalities in their suggestions/advice/observations/recommendations?

5. Has any particular digital device been tested in this work? What is the outcome of the study?

6. PLOS authors have the option to publish the peer review history of their article (what does this mean?). If published, this will include your full peer review and any attached files.

**Do you want your identity to be public for this peer review?** For information about this choice, including consent withdrawal, please see our Privacy Policy.

Reviewer #1: Yes: Laritza M Rodriguez MD, PhD

Reviewer #2: Yes: Towhida Ahsan

Reviewer #3: Yes: Subhagata Chattopadhyay

---

## [Decision Letter · Decision Letter 1]

23 Oct 2023

PDIG-D-23-00250R1

Experts’ views on translating NHS support to stop smoking in pregnancy into a comprehensive digital intervention

PLOS Digital Health

Dear Dr. McDaid,

Thank you for submitting your manuscript to PLOS Digital Health. After careful consideration, we feel that it has merit but does not fully meet PLOS Digital Health's publication criteria as it currently stands. Therefore, we invite you to submit a revised version of the manuscript that addresses the points raised during the review process.

Please submit your revised manuscript within 30 days Nov 22 2023 11:59PM. If you will need more time than this to complete your revisions, please reply to this message or contact the journal office at digitalhealth@plos.org. Please include the following items when submitting your revised manuscript:

We look forward to receiving your revised manuscript.

Kind regards,

Haleh Ayatollahi

Section Editor

PLOS Digital Health

Journal Requirements:

a. State what role the funders took in the study. If the funders had no role in your study, please state: “The funders had no role in study design, data collection and analysis, decision to publish, or preparation of the manuscript.”

b. If any authors received a salary from any of your funders, please state which authors and which funders.

Additional Editor Comments (if provided):

Reviewers' comments:

Reviewer's Responses to Questions

**Comments to the Author**

1. If the authors have adequately addressed your comments raised in a previous round of review and you feel that this manuscript is now acceptable for publication, you may indicate that here to bypass the “Comments to the Author” section, enter your conflict of interest statement in the “Confidential to Editor” section, and submit your "Accept" recommendation.

Reviewer #1: All comments have been addressed

Reviewer #2: (No Response)

Reviewer #3: All comments have been addressed

2. Does this manuscript meet PLOS Digital Health’s publication criteria? Is the manuscript technically sound, and do the data support the conclusions? The manuscript must describe methodologically and ethically rigorous research with conclusions that are appropriately drawn based on the data presented.

Reviewer #1: Yes

Reviewer #2: (No Response)

Reviewer #3: Yes

3. Has the statistical analysis been performed appropriately and rigorously?

Reviewer #1: Yes

Reviewer #2: (No Response)

Reviewer #3: N/A

4. Have the authors made all data underlying the findings in their manuscript fully available (please refer to the Data Availability Statement at the start of the manuscript PDF file)?

Reviewer #1: Yes

Reviewer #2: (No Response)

Reviewer #3: (No Response)

5. Is the manuscript presented in an intelligible fashion and written in standard English?

Reviewer #1: Yes

Reviewer #2: (No Response)

Reviewer #3: Yes

6. Review Comments to the Author

Reviewer #1: Thank you for your submission.

Reviewer #2: We would like to express our gratitude to the authors for their outstanding efforts and for taking our recommendation into consideration. 

A recent publication in the journal TOXICS has concluded that there is evidence to suggest that the use of e-cigarettes during pregnancy could potentially harm the health of both the mother and the fetus. The exposure to e-cigarettes is associated with various adverse effects such as increased systemic inflammation, low birth weight, preterm birth, and small size for gestational age status. 

Ref: Vilcassim, M J Ruzmyn, et al. “Electronic Cigarette Use during Pregnancy: Is It Harmful?.” Toxics vol. 11,3 278. 18 Mar. 2023, doi:10.3390/toxics11030278

So, it is important to mention the harmful effects of electronic cigarettes (e-cigarettes) during pregnancy in conclusion for mass awareness and avoiding addiction to e-cigarettes instead of cigarette smoking.

Reviewer #3: While researching on pregnant smokers, I feel, understanding and giving a summary of neurobiology of smoking would help readers understanding the perspective in a better way. Researches have shown that pregnant women especially who are single, divorced, and separated (which are the SDoH parameters) smoke more. The key reason is to control their mood dysregulations during this phase of life. Therefore, I recommend authors to add a small section of 'Neurobiology of smoking'. I'm sure it'll strengthen the rationale of this work. Otherwise, I am happy with the explanations provided by the authors.

7. PLOS authors have the option to publish the peer review history of their article (what does this mean?). If published, this will include your full peer review and any attached files.

**Do you want your identity to be public for this peer review?** For information about this choice, including consent withdrawal, please see our Privacy Policy. 

Reviewer #1: Yes: Laritza M. Rodriguez MD, PhD

Reviewer #2: Yes: Towhida Ahsan

Reviewer #3: Yes: Subhagata Chattopadhyay

---

## [Decision Letter · Decision Letter 2]

19 Dec 2023

PDIG-D-23-00250R2

Experts’ views on translating NHS support to stop smoking in pregnancy into a comprehensive digital intervention

PLOS Digital Health

Dear Dr. McDaid,

Thank you for submitting your manuscript to PLOS Digital Health. After careful consideration, we feel that it has merit but does not fully meet PLOS Digital Health's publication criteria as it currently stands. Therefore, we invite you to submit a revised version of the manuscript that addresses the points raised during the review process.

Please submit your revised manuscript within 60 days Feb 17 2024 11:59PM. If you will need more time than this to complete your revisions, please reply to this message or contact the journal office at digitalhealth@plos.org. Please include the following items when submitting your revised manuscript:

We look forward to receiving your revised manuscript.

Kind regards,

Haleh Ayatollahi

Section Editor

PLOS Digital Health

Journal Requirements:

Additional Editor Comments (if provided):

Reviewers' comments:

Reviewer's Responses to Questions

**Comments to the Author**

1. If the authors have adequately addressed your comments raised in a previous round of review and you feel that this manuscript is now acceptable for publication, you may indicate that here to bypass the “Comments to the Author” section, enter your conflict of interest statement in the “Confidential to Editor” section, and submit your "Accept" recommendation.

Reviewer #1: All comments have been addressed

Reviewer #2: (No Response)

Reviewer #3: (No Response)

2. Does this manuscript meet PLOS Digital Health’s publication criteria? Is the manuscript technically sound, and do the data support the conclusions? The manuscript must describe methodologically and ethically rigorous research with conclusions that are appropriately drawn based on the data presented.

Reviewer #1: Yes

Reviewer #2: Yes

Reviewer #3: (No Response)

3. Has the statistical analysis been performed appropriately and rigorously?

Reviewer #1: Yes

Reviewer #2: N/A

Reviewer #3: (No Response)

4. Have the authors made all data underlying the findings in their manuscript fully available (please refer to the Data Availability Statement at the start of the manuscript PDF file)?

Reviewer #1: Yes

Reviewer #2: (No Response)

Reviewer #3: (No Response)

5. Is the manuscript presented in an intelligible fashion and written in standard English?

Reviewer #1: Yes

Reviewer #2: (No Response)

Reviewer #3: (No Response)

6. Review Comments to the Author

Reviewer #1: Thank you for your submission. Looking forward to follow-up work.

Reviewer #2: (No Response)

Reviewer #3: Please bring the SDoH parameters connected to neurobiology behind smoking in pregnancy and let the readers know how the issue could be managed by digitized interventions that the article is proposing. If not considered, I believe, any intervention will be highly inadequate and the end goal can not be achieved. Thank you.

7. PLOS authors have the option to publish the peer review history of their article (what does this mean?). If published, this will include your full peer review and any attached files.

**Do you want your identity to be public for this peer review?** For information about this choice, including consent withdrawal, please see our Privacy Policy. 

Reviewer #1: Yes: Laritza M Rodriguez

Reviewer #2: None

Reviewer #3: Yes: Subhagata Chattopadhyay

---

## [Decision Letter · Decision Letter 3]

16 Jan 2024

PDIG-D-23-00250R3

Experts’ views on translating NHS support to stop smoking in pregnancy into a comprehensive digital intervention

PLOS Digital Health

Dear Dr. McDaid,

Thank you for submitting your manuscript to PLOS Digital Health. After careful consideration, we feel that it has merit but does not fully meet PLOS Digital Health's publication criteria as it currently stands. Therefore, we invite you to submit a revised version of the manuscript that addresses the points raised during the review process.

Please submit your revised manuscript within 60 days Mar 16 2024 11:59PM. If you will need more time than this to complete your revisions, please reply to this message or contact the journal office at digitalhealth@plos.org. Please include the following items when submitting your revised manuscript:

We look forward to receiving your revised manuscript.

Kind regards,

Haleh Ayatollahi

Section Editor

PLOS Digital Health

Journal Requirements:

Additional Editor Comments (if provided):

Thanks for your time and efforts to revise the manuscript. However, it seems that the reviewer is not satisfied with the revision. I appreciate if you please carefully address the reviewer's concerns.

Reviewers' comments:

Reviewer's Responses to Questions

**Comments to the Author**

1. If the authors have adequately addressed your comments raised in a previous round of review and you feel that this manuscript is now acceptable for publication, you may indicate that here to bypass the “Comments to the Author” section, enter your conflict of interest statement in the “Confidential to Editor” section, and submit your "Accept" recommendation.

Reviewer #3: (No Response)

2. Does this manuscript meet PLOS Digital Health’s publication criteria? Is the manuscript technically sound, and do the data support the conclusions? The manuscript must describe methodologically and ethically rigorous research with conclusions that are appropriately drawn based on the data presented.

Reviewer #3: (No Response)

3. Has the statistical analysis been performed appropriately and rigorously?

Reviewer #3: (No Response)

4. Have the authors made all data underlying the findings in their manuscript fully available (please refer to the Data Availability Statement at the start of the manuscript PDF file)?

Reviewer #3: (No Response)

5. Is the manuscript presented in an intelligible fashion and written in standard English?

Reviewer #3: (No Response)

6. Review Comments to the Author

Reviewer #3: Dear authors, I'm still sticking to my point as a reviewer. It has not been addressed in the revised manuscript. To me, it is therefore inadequate for publication. However, I leave it to the editor of PLOS Digital Health to take the decision.

7. PLOS authors have the option to publish the peer review history of their article (what does this mean?). If published, this will include your full peer review and any attached files.

**Do you want your identity to be public for this peer review?** For information about this choice, including consent withdrawal, please see our Privacy Policy. 

Reviewer #3: Yes: Subhagata Chattopadhyay

---

## [Editor Report · Decision Letter 4]

17 Feb 2024

Experts’ views on translating NHS support to stop smoking in pregnancy into a comprehensive digital intervention

PDIG-D-23-00250R4

Dear Dr. McDaid,

We are pleased to inform you that your manuscript 'Experts’ views on translating NHS support to stop smoking in pregnancy into a comprehensive digital intervention' has been provisionally accepted for publication in PLOS Digital Health.

Best regards,

Haleh Ayatollahi

Section Editor

PLOS Digital Health